# Directional Submicrofiber Hydrogel Composite Scaffolds Supporting Neuron Differentiation and Enabling Neurite Alignment

**DOI:** 10.3390/ijms231911525

**Published:** 2022-09-29

**Authors:** Lena Mungenast, Fabian Züger, Jasmin Selvi, Ana Bela Faia-Torres, Jürgen Rühe, Laura Suter-Dick, Maurizio R. Gullo

**Affiliations:** 1Institute for Chemistry and Bioanalytics, University of Applied Sciences FHNW, Hofackerstrasse 30, 4132 Muttenz, Switzerland; 2Institute for Medical Engineering and Medical Informatics, University of Applied Sciences FHNW, Hofackerstrasse 30, 4132 Muttenz, Switzerland; 3Department of Microsystems Engineering, University of Freiburg–IMTEK, Georges-Koehler-Allee 103, 79110 Freiburg, Germany

**Keywords:** neural cell guiding, neurite alignment, electrospinning, fiber-hydrogel scaffold

## Abstract

Cell cultures aiming at tissue regeneration benefit from scaffolds with physiologically relevant elastic moduli to optimally trigger cell attachment, proliferation and promote differentiation, guidance and tissue maturation. Complex scaffolds designed with guiding cues can mimic the anisotropic nature of neural tissues, such as spinal cord or brain, and recall the ability of human neural progenitor cells to differentiate and align. This work introduces a cost-efficient gelatin-based submicron patterned hydrogel–fiber composite with tuned stiffness, able to support cell attachment, differentiation and alignment of neurons derived from human progenitor cells. The enzymatically crosslinked gelatin-based hydrogels were generated with stiffnesses from 8 to 80 kPa, onto which poly(ε-caprolactone) (PCL) alignment cues were electrospun such that the fibers had a preferential alignment. The fiber–hydrogel composites with a modulus of about 20 kPa showed the strongest cell attachment and highest cell proliferation, rendering them an ideal differentiation support. Differentiated neurons aligned and bundled their neurites along the aligned PCL filaments, which is unique to this cell type on a fiber–hydrogel composite. This novel scaffold relies on robust and inexpensive technology and is suitable for neural tissue engineering where directional neuron alignment is required, such as in the spinal cord.

## 1. Introduction

Tissue engineering for regenerative medicine aims to achieve the physiological performance of engineered tissue by mimicking the native architecture. Scaffolds are often used to bridge tissue loss and support host and proliferating cells in laying down new extracellular matrix, necessary for the formation of functional tissue structures [1]. Anisotropic native neural tissues, such as spinal cord or brain, have been shown to require a sophisticated scaffold design for cell attachment, proliferation, differentiation and guidance [1,2,3]. Approaches for efficient cell guidance and alignment often include (bio)chemical cues combined with a topography consisting of structures such as microgrooves, channels or nano-and microfibers [1]. The microfibers’ resemblance to the three-dimensional microstructure of extracellular matrix (ECM) can be accomplished by electrospinning of polymers—a simple, versatile and cost-effective technology. Electrospun fiber networks are characterized by high porosity, large surface area, submicron diameter and strong interconnection of the fiber, thereby closely mimicking the 3D fiber organization of the natural extracellular matrix, in particular that of collagens, ranging from single collagen fibrils of 20–500 nm to collagen bundles of several microns [4,5,6]. To engineer hierarchically structured scaffolds, the controlled deposition of fibers is crucial and several techniques for the alignment of electrospun fibers have been developed, including a rotating drum [7,8,9], micro patterned collectors [10], electrospinning and photolithography [11] or near-field electrospinning [12]. It has been shown that axonal alignment of neurons can be triggered and neurite outgrowth can be promoted on submicron electrospun poly(ε-caprolactone) (PCL) fiber scaffolds [9,13,14,15,16,17]. PCL has been shown to be biocompatible, suitable for electrospinning, capable to guide neural cells in vitro, and is approved by the FDA [18,19]. In addition to topographical features, the design of microenvironment mimicking native tissue properties is of particular importance in supporting the differentiation of (human neural) progenitor cells [1,20,21]. Since the tissue microenvironment is also determined by the stiffness, influencing many cellular processes such as adhesion, proliferation and differentiation [20,22], hydrogels can be utilized to adapt the stiffness of cell culture substrates. The elastic modulus for spinal cord tissue is considered one of the lowest found in the human body and is considered relevant below 100 kPa [20,23,24,25,26]. Mismatching the stiffness of a scaffold/implant and that of the host tissue will alter the migration behavior and differentiation cycle of neural progenitor cells into neurons [18,25,26,27,28]. The most commonly used hydrogels in neural tissue engineering, hyaluronic acid and poly (ethylene glycol), require crosslinking usually involving the use of expensive, toxic chemicals or time-consuming pre-modification of the monomer to enable UV-crosslinking, which in turns makes it more challenging to use those hydrogels for cell encapsulated hydrogel scaffolds [27,28,29,30]. Due to their availability, good biocompatibility and low cost, gelatin-based hydrogels have often been used as cell culture substrates [31]; however, thermally crosslinked gelatin hydrogels have low stability and a high degradation rate at physiological conditions [32]. To provide mechanical and proteolytic stability, gelatin hydrogels enzymatically crosslinked with transglutaminase (TG) offer a low cost, fast, environmentally friendly, non-toxic alternative for gelatin hydrogels. By varying the gelatin and enzyme concentrations/ratios, the elastic modulus of the hydrogels can be tuned to achieve the desired tissue-relevant stiffness. Gelatin-methacryloyl (GelMa) hydrogels with 35 kPA elastic modulus have been proven to support adhesion and proliferation of PC12 neuronal model cells, and to further enable cell spreading and neurite outgrowth [33,34]. TG-crosslinked gelatin hydrogels patterned by microcontact printing enabled the adhesion and proliferation of Schwann cells while maintaining healthy morphology, adhesion and extensions of axons [35].

In this work, we present the combination of topographical and microenvironmental features leading to a novel gelatin-based hydrogel–fiber composite scaffold tunable in stiffness, fiber alignment and fiber density, specifically adjusted to support the attachment, differentiation and alignment of neurons derived from a human progenitor cell line (ReN VM cells), capable to differentiate into neurons, astrocytes and oligodendrocytes. While all these cells are of utmost importance and key players of the central nervous system, neurons remain the main cell type responsible for signal transmission from the brain to the peripheral nervous system. Combining the benefits of tunable hydrogel stiffness and electrospun microfibers into a hybrid cell culture substrate, we produced a scaffold with tissue-relevant elastic modulus and topographical guidance cues that support neuronal differentiation and cellular alignment. Such a composite material can be developed as an implantable scaffold to support the regeneration of spinal cord in patients after traumatic injury. Moreover, scaffolds featuring directionality and alignment cues may also be implemented as in vitro neuronal models and serve the investigation of therapies’ efficacy promoting neural regeneration.

## 2. Results

For the hydrogel–fiber composite serving as the scaffold for the culture of neural cells, gelatin was selected as the base material for the hydrogel with stiffnesses ideal for the support of neural cells. As illustrated in Figure 1, for the second layer, located on top of the hydrogel, a sparse layer of electrospun fibers representing the topographical cue was deposited. For the generation of the fibers, the synthetic polymer poly-ε-caprolactone was chosen. This polymer was used to produce a fiber layer which was thin enough so that the gelatin was not covered completely. The combination of these two starting materials allowed the production of hydrogel–fiber scaffolds, which were characterized using optical microscopy and nanoindentation, prior to application in in-vitro cell culture.

Structural support of culture, viability and differentiation of human neural progenitor cell lines is provided by customized hydrogels. A key feature of these gels is to match the Young’s Modulus of the cells of the central nervous system, which is in the typical stiffness range 20 to 100 kPa. To this purpose, aiming to gradually increase the elastic moduli, three hydrogels with different concentrations of gelatin (10%, 15%, 20% *w*/*w*), were produced. For crosslinking, transglutaminase with a concentration of 50 mg/mL was added to the gelatin solution and incubated for 12 h at 37 °C. In this process, acyl transfer of a lysine bond leads to intra- and intermolecular crosslinking of proteins through formation of an isopeptide bond. The average sample thickness was 2.0 mm. As shown in Figure 2a, at a temperature of 37 °C the stiffness of all hydrogels (6.8 kPa ± 0.5; 20.7 kPa ± 0.4; 79.0 kPa ± 0.8) was within the desired range (20–100 kPa) and increases with increasing gelatin concentration. In comparison to this, the elastic modulus of a standard tissue-culture polystyrene flask is around 3.5 GPa [36].

To evaluate the effect of hydrogel stiffness on cell adhesion and proliferation, the progenitor ReN VM cells were seeded on plain gelatin hydrogels with the three different elastic moduli (7 kPa, 21 kPa, 80 kPa) described above. In addition, the culture on hydrogels without topographical cues served as a positive control to investigate the influence of topographical alignment cues in later experiments. All three hydrogel concentrations enabled the adhesion and proliferation of the cells, as determined by the increased number of viable cells and depicted in the optical micrographs in Figure 2b,c, respectively. The cells displayed a spread morphology especially in the case of the lower stiffness hydrogels, indicating that the laminin-coated hydrogels provided sufficient cell-anchoring points on the surface. The cell viability even after two weeks of culture, as evidenced by the increased cell numbers on the hydrogel, as well as the spread morphology of the cells, are strong indicators of the biocompatibility of the scaffold. Moreover, based on the Alamar blue assay, the number of viable cells on all tested hydrogels was significantly higher compared to the reference material (laminin coated tissue-culture polystyrene: TCPS) at day 14, which demonstrates the superior cell supporting nature of the engineered hydrogels. As no significant differences in cell viability could be found between 7 and 21 kPa hydrogels, the latter was selected for the creation of the hybrid fiber–hydrogel structures.

As expected, the plain hydrogels could not provide sufficient alignment cues for neuronal cells. The ReN VM cells did not spontaneously grow with a preferred directionality but were randomly distributed on the hydrogel scaffold (Figure 2c). To promote the emergence of a directional alignment, directionally aligned PCL fibers and randomly aligned fibers as a control were spun directly on top of the hydrogels by electrospinning (Figure 3a). The electrospinning process was carried out such that only very thin mono layers (<1 µm) of oriented fibers were produced. The degree of alignment was assessed by the orientation of the fibers along the reference axis at 0° which represented the axis along which the most fibers were oriented. By optimizing the spinning parameters with focus on polymer flow rate–needle-collector distance–deposition time, the fiber density for each deposition was adjusted to 25 ± 3.0% fiber coverage on the hydrogel. This enabled a regular mean spacing of 20 ± 5 µm between the fibers’ monolayer (Figure 3c), and a fiber thickness in the micron range. The fiber diameter obtained was in the sub-micrometer range for both random (0.86 ± 0.08 µm) and aligned (0.84 ± 0.06 µm) fibers. These parameters were chosen so as to offer sufficient topographical cell guidance cues without altering the substrate stiffness or acting as a barrier to cell movement by overloading the hydrogel with fibers. As depicted in Figure 3b, the volume view of the fiber–hydrogel scaffold displays the 3D organization of the fibers on top of the hydrogel. The hydrogel serves as the base material and presents an even surface for the fibers. In this three-dimensional view, the deposition of the fibers on the surface without significant embedding into the hydrogel is shown. The cross-sectional view clearly indicates that the fibers are deposited as a mono layer on the top of the hydrogel surface, offering individual topographical cues with an average height of 860 ± 80 nm above the surface. The resulting fiber layer did not induce any significant difference on the elastic modulus of plain hydrogels (Figure 3d), thus maintaining the tissue-relevant stiffness.

The deposition of the fibers in the copper tube installation led to straight fibers deposited in a highly parallel fashion; here shown in Figure 3a, the fibers are aligned along the main axis horizontally. As depicted in the histogram in Figure 3c, the fiber deposition had a very narrow distribution of directionality with >90% of fibers distributed within ±10°. In comparison, the fibers deposited randomly showed a slightly coiled structure, often found for randomly distributed electrospun fibers, and they were deposited without a significant main axis. When a sample with random deposition is taken and the angle with highest fiber deposition is viewed (here −22°), only 15% of fibers are distributed within ±10° of this angle. To ensure the integrity of the fiber–hydrogel constructs during the experimental period, their long-term stability was tested in cell culture medium at physiological temperature (37 °C) for 28 days. The condition of the incubated fiber–hydrogel constructs was assessed by optical micrographs (Figure 3e, 28 days-time point). Although the fibers were not chemically bound to the surface of the hydrogel, neither delamination of fibers from the hydrogel, nor PCL degradation nor loss of hydrogel integrity were observed.

As established in the previous experiments, the hydrogels supported the adhesion and proliferation of neural progenitor cells; however, these plain hydrogels were not capable to guide and orient neural cells. To investigate the influence of topographical cues on the behavior of neural progenitor cells and committed neurons, ReN VM cells were cultured on hydrogels containing random and aligned electrospun PLC fibers and compared to the plain hydrogels. Cell viability assays showed no negative effect of the fibers on the cells’ metabolic activity and cell numbers within the 14 days culture period. As shown in Figure 4a, proliferation of ReN cells was similar on aligned fibers on hydrogel, random fibers on hydrogel and on plain hydrogel scaffolds.

During the differentiation of ReN VM cells to neurons, the cell viability on all tested scaffolds plateaued from day 7 to day 21 (Figure 4b). This stagnation of cell proliferation, combined with the fact that the optical micrographs (Figure 5a) indicate healthy and adherent cells, suggests the initiation of cell differentiation process. The differentiation of progenitor cells was confirmed by immunostaining of specific marker at day 1 and 14 (additional images in Appendix A). Immunofluorescence revealed a very low expression level of the neural stem cell marker nestin, and expression of tubulin and choline acetyl-transferase (ChAT), hallmarks of neural differentiation in all scaffolds (Figure 5b). In addition, the elongation and alignment of neurites was evident on hybrid scaffolds (Figure 5a,b). The intensity of expression of tubulin is comparable for both fiber–hydrogel scaffolds, as well as the plain hydrogel (Figure 5d), which indicates a similar efficacy of fibers and hydrogel at the induction of the differentiation process of human neural progenitor cells into neurons. Notably, the expression of ChAT showed to be higher in the aligned scaffold, pointing towards a preferential motor-neuron like differentiation in the scaffolds with aligned fibers, thereby implicating a potential influence of the topography on the differentiation pathway.

Nevertheless, hybrid fiber–hydrogel scaffolds showed themselves to be essential for in vivo-like directional alignment of the differentiated neurons (Figure 5a–c), visible after the 14th day of cell culture. More than 50% of the cells were positioned within an angle of deviation of <10% with regard to the main axis of the fiber scaffold.

The differentiated neurons clearly followed the fiber pattern on the hydrogel and built a well oriented network at day 14 for both random and aligned fibers (Figure 5a). Neuronal bodies were slightly elongated, and neurites extended (Figure 5b). The narrower distribution of neurons on the aligned PLC fiber scaffolds, in comparison to that on random fibers or hydrogel (Figure 5c) where the neurons built completely random networks, suggests that topographically aligned cues were essential for neuronal orientation. Additionally, “nerve bundle”-like arrangements, only observed along the aligned PCL fibers, imply not only that this fiber–hydrogel scaffold influences the directionality of individual neurons and their neurites, but also that it enables tissue-like association of neurons (Figure 5e). This was particularly observed when the space between PCL topographical cues was extended and at longer cultivation periods (post day 14).

## 3. Discussion

Neural tissues, including the spinal cord, are anisotropic and complex tissues. Scaffolds that provide biochemical and topographical cues resembling those seen in the native tissues strongly support tissue engineering and regenerative medicine [37]. Previous studies focusing on neural progenitor cell supporting scaffolds reported the importance of hydrogel properties such as elastic modulus on their differentiation success and the suitability of topographical cues for the successful alignment of neural cells [17,38,39]. For many implemented neuronal models, the used scaffolds either lack topographical cues [40,41,42] to replicate the anisotropic nature of spinal cord, or the stiffness for tissue-relevant substrates [16,43]. Our composite fiber–hydrogel scaffolds provide the appropriate stiffness as well as alignment cues to support both the differentiation of the neural progenitor cell line ReN cell VM into neurons and their directional alignment. The human progenitor cell line used here can be differentiated simultaneously or selectively into neurons, astrocytes and oligodendrocytes, avoiding the use of expensive, labor-intensive and limited primary cells. The successful integration with the here presented fiber–hydrogel scaffolds will be beneficial for cell co-cultures assays in further studies.

The fabrication of fiber–hydrogel composite scaffolds was successfully achieved by producing crosslinked gelatin hydrogels which served as the collection substrate for electrospun PCL fibers. Furthermore, the mechanical stability of the 21 kPa hydrogel would enable convenient handling for surgeons in a clinical setting. The stiffness of our gelatin-based hydrogels was in the range of that typically observed for low percentage porcine Type A gelatin crosslinked by TG and other comparable GelMa based hydrogels [33,35]. Likewise, with PCL fibers added, our composite scaffolds showed the elastic modulus of neural tissue, e.g., spinal cord with 20–100 kPA [23,24,25,44], and provided a suitable substrate for cell proliferation and differentiation of human progenitor cells. The additional laminin coating guaranteed cell adhesion since, in non-coated scaffolds, neural cells tend to detach very fast due to the lack of adhesion motifs [35,45]. For anisotropic scaffold design, the PCL fiber deposition was performed in an aligned fashion directly on the hydrogel. The typical electrospinning-engineered submicron PCL fiber diameter was achieved for aligned as well as randomly distributed fibers. Our homemade gap-based electrospinning setup yielded highly parallel fiber distribution, comparable to alignment distributions achieved using a rotating mandrel [46,47], with the advantage of being amenable to a broad range of substrates as collectors. In comparison, the use of hydrogels in a rotating drum is typically challenging due to (1) the limited mechanical stability of hydrogels when used as substrates upon the high rotation speed of the drum required for sufficient alignment, and (2) the difficult transfer of a fragile fiber mesh without destruction of the fibers themselves, and of the generated aligned pattern onto the designated hydrogel, when fibers are spun on a collector plate and not directly on the hydrogel [48,49]. The technical challenge of depositing highly aligned fibers on a not perfectly flat surface, such as that of a hydrogel, was overcome by optimizing the needle-collector distance, placement of collection substrate and polymer flow rate. This resulted in PCL fiber–hydrogel scaffolds with more than 90% of fibers within ±12°, superior to those found on 3D collectors substrates achieving more than 80% of fibers within ±20° deviation to a reference line [9,50]. The topographical sub-micrometer cues provided by the engineered scaffolds were in the same range as observed in collagen fibers (20–500 nm) [6] of the native extracellular matrix and acted as a trigger of in vivo-like cell alignment [51,52]. Our fabrication process also allowed control of the fiber density in order to avoid highly dense fiber meshes that can hinder the contact of cells to the underlying hydrogel substrate and therefore diminish the positive influence of in vivo-relevant hydrogel stiffness [46,53]. Hence, with the optimized setting, we can manipulate the balance between number of topographical cues and inter-fiber spacing on the hydrogel, an advantage for the production of micro topographies by molding or hot embossing where a new expensive mask is required for every new casting. The tuning of alignment sites in our scaffolds is leading on the one hand to an alignment of individual neurites and, on the other hand, to an aligned growth of several neurons. The individual neurite growth clearly follows the undulating shape of the PCL fibers (Figure 5a,b) and is therefore a clear indication of topography being the main alignment cue. In addition to the topographical cues, the aligned neuronal and neuron bundle growth could also be triggered by an anisotropic elasticity in the hydrogel, eventually induced by the deposition of aligned PCL fibers [54,55,56]. Further detailed local elasticity measurements would, however, be necessary to reveal the presence of an elastic anisotropy in the hydrogel and assess its mechanotransduction pathways in the alignment process. The results presented here confirm the importance of scaffold design as previously shown for in vivo PCL-based scaffolds by Wong et al. [21], indicating that the association and orientation of neurons into “nerve bundles” for successful spinal cord regeneration is influenced by topographical submicron cues. The biocompatibility of our fiber–hydrogel scaffold for ReN VM cells is in accordance with plain gelatin-based scaffolds [35] but offers the additional feature of topographical cues. When we compare cells on scaffolds based on random PCL fibers on glass, which have been shown to promote differentiation but no alignment [16], our scaffolds led to a comparable expression profile of neuronal differentiation markers, in addition to the successful spatial alignment. The alignment of cells along the fibers, quantified by the deviation angle of the cells’ main axis to the fiber scaffold, is similar to PCL scaffolds previously reported in the literature for other neural progenitor cell lines [46,57]. The differentiation of neural progenitor cells into neurons on the scaffold was verified by confocal imaging of tubulin ß III, an early differentiation marker in the cytoskeleton. Significant tubulin ß III expression was observed from day 14 on. These results are in accordance with previously published data showing that the differentiation of ReN VM cells leads to significantly increased levels of tubulin ß III in the neural cell population over 14 days [16,58]. Choline acetyltransferase (ChAT), the enzyme producing the neurotransmitter acetylcholine in motoneurons [59], was predominately expressed from day 7 on in the differentiated ReN VM cells, indicating that they preferentially followed a motoneuronal differentiation pathway. This was particularly observed in the aligned scaffolds, implicating a potential influence of the topography on the differentiation pathway. This is of particular interest in tissue engineering due to the demand for models of traumatic injuries like spinal cord injury, since the intrinsic regrowth ability of motoneurons is higher than for sensory neurons, and thereby offers an attractive target to regenerate functional circuits within the central nervous system [60].

Here, and for the first time, the neural progenitor ReN VM cell line was differentiated into directionally aligned motor neuron-like cells on an optimized fiber–hydrogel scaffold, displaying tissue-relevant stiffness and oriented fibers serving as topographical cues for cell alignment. This novel approach for scaffold manufacturing provides an optimized set of cues that allows differentiation of the neural progenitor cells in vitro.

## 4. Materials and Methods

### 4.1. Production of Gelatin Hydrogels

Gelatin (from porcine skin, Sigma G1890, St. Louis, MI, USA) was dissolved at concentrations of 10%, 15% and 20% *w*/*w* in PBS (Sigma, 806552) and the pH adjusted to 7.4. Transglutaminase (Ajinomoto) with a concentration of 50 mg/mL was added 1/9 *v*/*v* to the gelatin solution and incubated for 12 h at 37 °C for enzymatic crosslinking. The gels were produced in a 24-well plate format with an average thickness of 2.0 mm ± 0.2 mm and stored in PBS at 4 °C until further use. All work involving cell culture was conducted under sterile conditions.

### 4.2. Fiber Spinning

Prior to electrospinning, polycaprolactone (PCL, Sigma, 440744) was dissolved as a 20% *w*/*w* solution in 30/70% *v*/*v* acetic acid/acetone (Sigma, 695092/Sigma, 34850). The electrospinning setup was custom made and included a high voltage generator (AIP Wild, FUG HCB), a syringe pump (TSE systems, 540080) and a copper collector plate. The following optimized electrospinning parameters were employed for the spinning solutions: 20 kV voltage, 15 cm distance from needle to collector, 1.0 mL/h polymer flow rate, and 1-5 min spinning time. For the aligned deposition of fibers, the distance between the copper rods, placed on either side of the collector glass slide, was between 3 and 5 cm. The spinning was performed on thin pre-crosslinked gelatin-transglutaminase hydrogels. For cell culture experiments, the fiber–hydrogels were punched out with biopsy punchers to the required size and sterilized (20 min UV) prior to use. All hydrogel fiber scaffolds were coated with 10 µg/mL laminin (Merck, L2020, Kenilworth, NJ, USA) for 2 h at 37 °C.

### 4.3. Characterization of Hydrogel–Fiber Scaffold

The stiffness of the hydrogels was determined using a nano indenter (Optics11, Piuma, Amsterdam, The Netherlands). Hydrogel samples, gelatin only and PCL-gelatin fiber–hydrogels, were both immersed in PBS and kept at 37 °C during the measurement. Indentations were performed using a cantilever (Optics11, P210388M), with geo factor 2.52 in air, 0.28 N/m spring constant and a spherical tip (r = 53 µm), in displacement mode. Each sample was indented in scan mode 3 × 3 with (dx, dy = 2000 µm) thrice. The data fitting and calculation of the elastic modulus was based on the Hertzian-contact model.

Optical micrographs were taken with an Olympus IX73 inverted microscope. The fiber diameter and alignment were analyzed on optical micrographs with ImageJ using the directionality plug-in. The “Fourier components” analysis method with number of bins Nbin = 90 from −90° to 90° was applied on 8-bit images and the data plotted as frequency distributions. Different fiber densities resulting from varying spinning parameters (time = 1 min, 3 min, 5 min; distance = 15, 20 cm) were determined from segmented bimodal images (N = 9) by using ImageJ [46,47].

Three-dimensional images of fiber–hydrogel composites were taken with a confocal microscope (Olympus, FV3000, Tokyo, Japan). The gelatin hydrogel was stained with 1/100 flamingo protein gel stain (Bio-Rad, 1610491, Hercules, CA, USA) and then used as base material for electrospinning. PCL was mixed with 1 µg/mL Nile red (Sigma, N31013) solution and used as previously described for electrospinning.

### 4.4. Neural Cell Culture

The human neural progenitor cell line ReN VM (Sigma Aldrich, SSC08) was cultured in proliferation medium composed of DMEM/F12 medium (Dulbecco’s modified Eagle’s medium/Nutrient mix F-12, Gibco, 11320, Waltham, MA, USA) supplemented with B27 supplement (Gibco, 17504044), Heparin (Stemcell technologies, 07980, Vancouver, Canada), EGF (Sino Biologics, 10605-HNAE, Beijing, China), bFGF (Thermo Fisher, PHG0024, Waltham, MA, USA) and 1.0% penicillin-streptomycin (Gibco, 15140122). For all in vitro experiments, cells at passage 10–20 were used. Throughout the cultivation period, the cells were incubated at 37 °C and 5% CO_2_ and the medium was exchanged every two or three days. Prior to seeding of ReN VM cells, cell culture flasks were coated with 10 µg/mL laminin (Merck, L2020, Kenilworth, NJ, USA) for 2 h at 37 °C to promote cell adhesion. To induce neuronal differentiation of cells on the scaffold, the medium was changed to specific neuronal differentiation medium composed of neurobasal medium (Gibco, 21103049) supplemented with B27 supplement (Gibco, 17504044), GlutaMax (Gibco, 35050038) and 1.0% penicillin streptomycin (Gibco, 15140122) at day 1 [61].

### 4.5. Characterization of Cell Adhesion, Proliferation and Differentiation

To characterize the biocompatibility of the scaffolds and study the cell behavior on the scaffolds, ReN cells were detached from the cell culture flask using Accutase (Gibco, A1110501) and then seeded onto the scaffolds with a density of 10,000 cells/scaffold. The adhesion of ReN VM cells on the scaffolds was analyzed by optical microscopy and images taken 24 h after seeding, with different magnifications. The proliferation of cells was determined by measuring the metabolic activity with an Alamar Blue assay (Thermo Fisher, DAL1025, Waltham, MA, USA). After 2 h of incubation, the fluorescence was measured at 560 nm with a plate reader (Agilent BioTek Synergy H4).

For immunofluorescence staining (after 1/7/14/21/28 days), cells were fixed at room temperature in 4% PFA solution (EM-grade, EMS, 15712), permeabilized with 0.2% Triton X-100 (Sigma, 94326) and blocked with 3% BSA (Sigma, A4503). After washing three times with PBS, the cells were incubated with the primary antibody solution overnight at 4 °C, followed by the secondary antibody solution at room temperature for 2 h. ReN VM cells were probed with the primary antibody anti-tubulin (Novus Bio, NB100 1612, 1/500, Centennial, CO, USA) with an anti-goat Alexa Fluor 555 secondary antibody (Abbexa, abx142427, 1/500, Cambridge, UK), anti-ChAT (Abcam, ab32454, 1/500, Cambridge, UK) with an anti-sheep Alexa Fluor 568 secondary antibody (Abcam, ab175712, 1/500) and anti-Nestin conjugated to Alexa Fluor 488 antibody (Stemcell, 60091AD, 1/500, Vancouver, Canada). After three washing steps with PBS, the cells were incubated at room temperature for 5 min with 4′6 diamidino-2-phenylindole (DAPI, Invitrogen, 1:1000, Waltham, MA, USA) to stain the nuclei. For imaging with a confocal microscope, the cells on hydrogels were mounted on a glass slide with antifade mounting medium (Thermo Fisher, P36982, Waltham, MA, USA). The quantification and intensity analysis of stained cells (Day 0, Day 14) were performed using CellProfiler [62], applying the ‘IdentifyPrimaryObjects’ and ‘MeasureObjectIntensity’ modules.

## 5. Conclusions

Enzymatically crosslinked gelatin-based hydrogels were generated with different stiffnesses, which successfully supported the adhesion, proliferation and differentiation of a human neural progenitor cell line. Utilizing electrospinning as a tool to add anisotropic features to the hydrogels, a sparse monolayer of polycaprolactone fibers was spun on top of the gel in either a random or aligned fashion without significantly altering the stiffness of the hydrogel substrate. On the fiber–hydrogel scaffolds, the progenitor ReN cells attached, proliferated and—after induction of differentiation—aligned their neurites along the polycaprolactone filaments. On these fiber–hydrogel scaffolds we could observe, for the first time using differentiating ReN cells, a directed organization at the single cell level (neurite extension), as well as at the cellular network level (“nerve bundles”).

The production of fiber–hydrogel scaffolds offers great potential for the fabrication of scaffolds for tissue engineering and regenerative medicine, combining the advantages of hydrogels (e.g., stiffness) with electrospun fibers replicating a native extracellular matrix and introducing topographical cues.

## Figures and Tables

**Figure 1 ijms-23-11525-f001:**
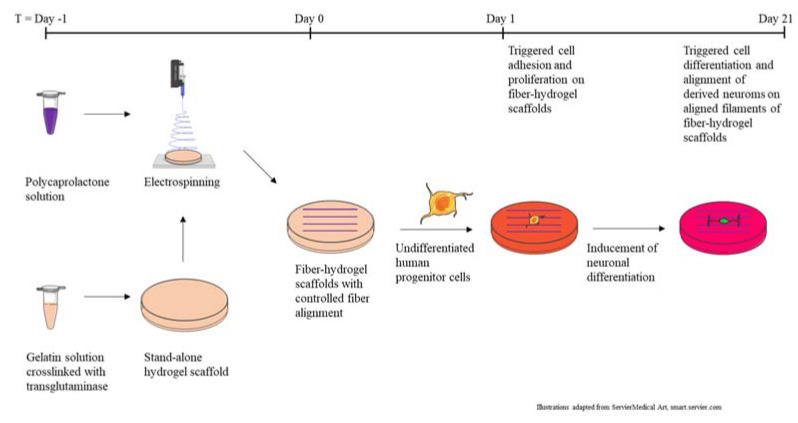
Production process of fiber–hydrogel scaffolds and use in cell culture experiments.

**Figure 2 ijms-23-11525-f002:**
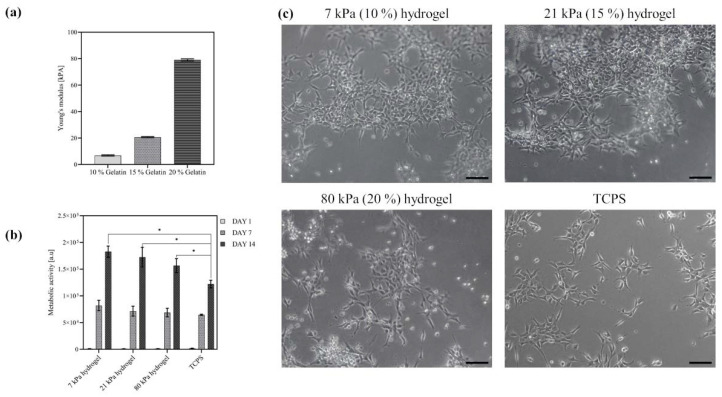
Gelatin hydrogels of variable stiffness support adhesion and proliferation of neural progenitor cells. (**a**) Elastic modulus [kPA] of transglutaminase-crosslinked gelatin hydrogels (10%, 15%, 20% *w*/*w*) at 37 °C, number of hydrogels N = 9; (**b**) Metabolic viability (determined by Alamar Blue assay) of ReN VM cells proliferated for 14 days on gelatin hydrogels; unpaired *t*-test (α < 0.05, N = 9); (**c**) Committed neurons at day 7 after differentiation from neural progenitor cells on 7 kPa, 21 kPa, 20 kPa gelatin hydrogels and TCPS control, scale bar = 100 µm. * *p* < 0.05.

**Figure 3 ijms-23-11525-f003:**
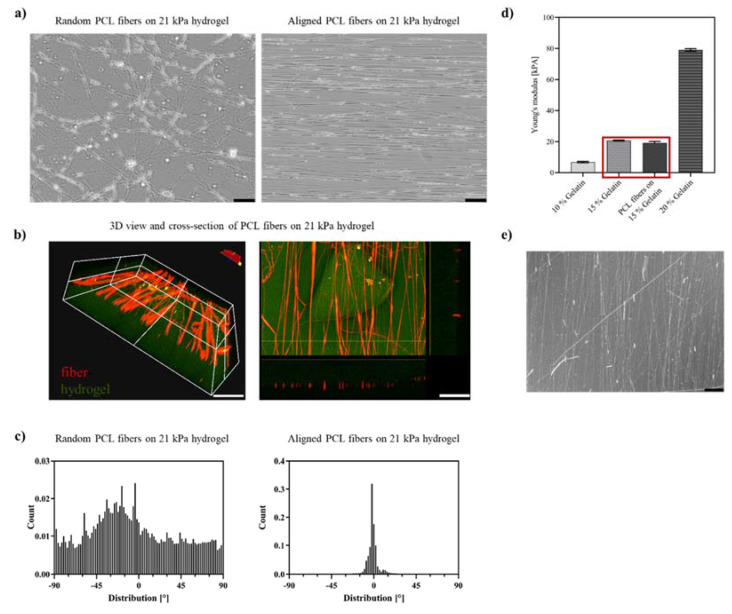
Scaffolds of electrospun PCL fibers on gelatin hydrogels. (**a**) Optical micrographs of random and aligned electrospun PCL fibers on 21 kPa hydrogels, scale bar = 100; (**b**) 3D volume view and cross section of confocal micrographs of PCL fibers (in red) on 21 kPa hydrogels (in green), scale bar = 200 µm; (**c**) Histograms with distribution of directionality µm of PCL fibers on 21 kPA hydrogels (left: random with number of fibers N = 200, right: aligned with number of fibers N = 500); (**d**) Elastic modulus [kPA] of transglutaminase-crosslinked gelatin hydrogels, highlighted with red box 15% gelatin with and without PCL fibers at 37 °C, number of hydrogels N = 9; (**e**) PCL fibers on 21 kPa gelatin hydrogel after 28 days at 37 °C/5% CO_2_ in cell culture medium, scale bar 100 µm.

**Figure 4 ijms-23-11525-f004:**
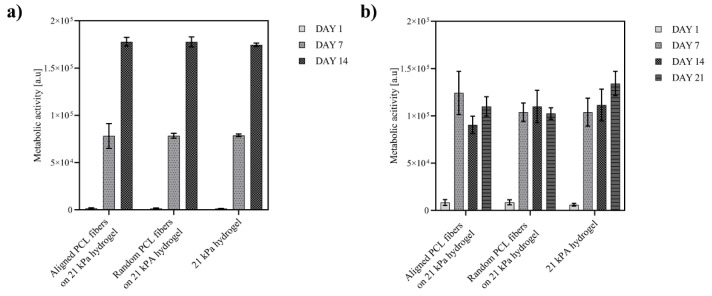
Neural progenitor cells and committed neurons on PCL-gelatin scaffolds. Cell viability (determined by Alamar Blue assay) of (**a**) ReN VM cells proliferated for 14 days and (**b**) committed neurons for 21 days on hydrogels and hybrid (aligned or randomly distributed) fibers on hydrogel scaffolds; number of hydrogels N = 9.

**Figure 5 ijms-23-11525-f005:**
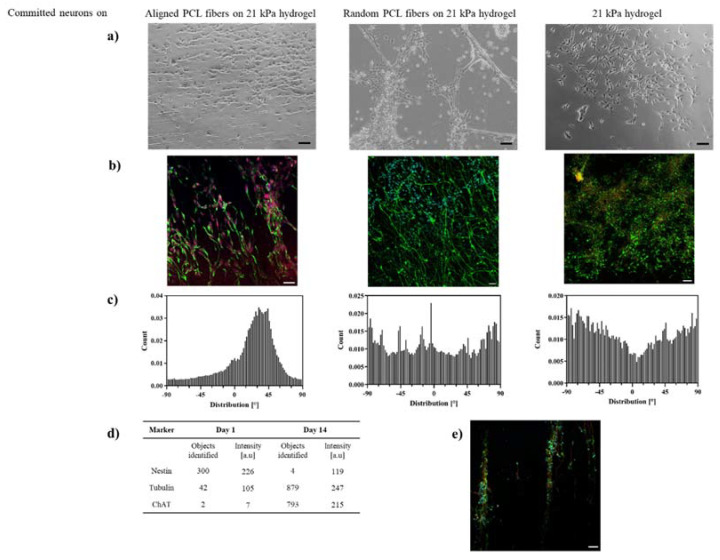
Alignment of committed neurons on PCL-gelatin scaffolds. (**a**) Optical micrographs and (**b**) confocal micrographs−immunostained with tubulin, nestin, ChAT and DAPI (green, yellow, red, blue) −of neurons on aligned, randomly distributed PCL-gelatin scaffolds and 21 kPa gelatin hydrogels at day 14; (**c**) Histograms of alignment of neurites on fiber-hydrogel scaffolds, number of cells N = 100; (**d**) Intensity analysis of nestin and tubulin of neurons on aligned PCL-gelatin scaffolds, number of images N = 9; (**e**) Neuro-bundle arrangement on aligned PCL-gelatin scaffolds, immunostained with tubulin, nestin, ChAT and DAPI (green, yellow, red, blue) at day 14, all scale bars = 100 µm.

## Data Availability

Data supporting reported results can be found at https://drive.switch.ch/index.php/apps/files/?dir=/Institution/Paper%20Neodent&fileid=4940792059 (accessed on 24 July 2022).

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
