# Peer review of "Directional Submicrofiber Hydrogel Composite Scaffolds Supporting Neuron Differentiation and Enabling Neurite Alignment"

_ijms, 2022, doi:10.3390/ijms231911525_

Round 1
Reviewer 1 Report
Reviewer’s comments
In this manuscript, the authors introduce their novel method to constitute a hybrid culture substrate which consists of a gelatin hydrogel with adjusted elasticity and on-top oriented PCL fibers intended to provide directional cue for cultured cells. The results showed that the differentiated neurons aligned with the PCL fibers and formed neurite bundles at some places, therefore, verified the endeavor of the study. The method and the results are interesting, and the manuscript was well written and organized.
Major comments
1. The authors stated that the aligned differentiated neurons resulted from the topographical cues of the hybrid substrate, but did not provide 3-dimensional topography of the substrate surface. The surface 3D topography is vital to verify the authors’ explanation for their results.
2. Relevant with the above comment, the results might be due to anisotropic stretch elasticity of the substrate surface induced by laying out the PCL fibers on the surface. Although the substrate elasticity may be characterized by indentation test, cultured cells on fibrous substrates actually stretch the surface to trigger the mechanotransduction pathways [1,2,3] (references listed below); therefore, if topping PCL fibers brought out anisotropic stretch elasticity to the substrate surface, it might induce the results as well. What do the authors make of this hypothesis.
3. The optimal elasticity of the substrate for neuronal differentiation (21 kPa) in this study is higher than that reported elsewhere [4], could the authors explain it?
4. The influence of the inter-fiber distance on the cellular behavior should be investigated.
5. Could the authors further explain how the neuronal differentiation on this hybrid substrate will be used clinically?
Minor comments
1. The number “1” in line 34 should be bracketed.
2. The first “a)” in line 199 should be deleted.
3. On Figure 5, the immunofluorescence staining for ChAT (red) and DAPI (blue) is not obvious, and the values with the scale bars on the pictures are obscure.
References
[1]. Trappmann B., Gautrot J.E., Connelly J.T., Strange D.G., Li Y., Oyen M.L., Cohen Stuart M.A., Boehm H., Li B., Vogel V., Spatz J.P., Watt F.M., Huck W.T. Extracellular-matrix tethering regulates stem-cell fate. Nat. Mater. 11, 642-9 (2012).
[2]. Wen J.H., Vincent L.G., Fuhrmann A., Choi Y.S., Hribar K.C., Taylor-Weiner H., Chen S., Engler A.J. Interplay of matrix stiffness and protein tethering in stem cell differentiation. Nat. Mater. 13, 979-87 (2014).
[3]. Feng Z, Fujita K, Yano M, Kosawada T, Sato D, Nakamura T, Umezu M. Physically-based structural modeling of a typical regenerative tissue analog bridges material macroscale continuum and cellular microscale discreteness and elucidates the hierarchical characteristics of cell-matrix interaction. Journal of the Mechanical Behavior of Biomedical Materials 126; 104956 (2022).
[4]. Engler A.J., Sen S., Sweeney H.L., Discher D.E. Matrix elasticity directs stem cell lineage specification. Cell 126, 677-89 (2006).
Reviewer 2 Report
The authors have conducted a study assessing the utility of their enzymatically-crosslinked gelatin hydrogels with pcl fibers for nerve guidance applications. They include strong rationale for their gel material and have chosen an appropriate cell line (ReN VM cells) to conduct initial assessments. Authors have a complete experimental design to assess the biocompatibility of their materials based on both mechanical properties and topographical cues. I support publishing of this well-written manuscript with clarification of key experimental details and statistical analysis as well as some addition of data to improve existing figures.
Introduction
-
Line 39 - please include a reference with scale of ecm to relate to your fiber diameter
-
Line 64 - 66 - Does reference 28 apply to the second half of the sentence? If not, please include a reference for the second half of the statement.
Results
-
Figure 1 - nice figure but please include the specific day when differentiation is induced - was this prior to seeding on scaffolds? This is a really important point as a premise of the paper is that the gels support neuron differentiation. However, it is unclear whether the differentiation is induced by the change of media - does the media change happen before or after cells are seeded on the hydrogels? Certainly the low kPa modulus supports neuronal differentiation, but it seems that in this case, the hydrogels support the adhesion and spreading of already differentiated neurons.
-
First two paragraphs read a bit like background information for introduction, particularly rationale for gelatin and pcl - consider moving to introduction
-
Line 115 - typos? “lysin” supposed to be lysine? “Intra- resp.” ?
-
Line 138 - please include a reference
-
Lines 139 - 143 - should be clear that gels also had laminin coating like ps control
-
Line 149 - incorrect figure reference?
-
Line 152 - define “very thin” layers (estimated thickness of fibers on surface?)
-
FIgure 3a - what are the bright, round “spots” in the random fiber image? They almost give the appearance of cells - are these bubbles? Some kind of aggregate? High intensity of overlapping fibers? Is this representative of all random fiber gels?
-
Figure 3d legend- remove “stability of” and include scale bar
-
FIgure 3 legend - there is a red box in 3c and red outline in 3b - the legend should indicate what those represent
-
Line 206 - 209 - please clarify time point
-
Figure 5c and d legend - how many cells analyzed?
-
Figure 5d - units for intensity
-
Figure 5e legend - which marker? Also taken day 14?
Important figure updates
-
Line 102/234/elsewhere - how much “space” between PCL (in microns)? for discussion of gelatin fiber density, figure 3 would benefit from analysis showing surface coverage as a percent of area of gel covered or number of fibers per given field of view width (especially as figure 3a fibers look to be a more sparse density in the random fiber image)
-
FIgure 5 - for fluorescent images, especially in first two, only tubulin is visible - manuscript would benefit from additional panels showing each channel and then the overlaid image to further support high tubulin and low nestin at day 14. Also include a row with earlier timepoint images to show reverse. Please include higher resolution images.
-
All figures with images - please make scale bar labels/ legends consistent (e.g. include distance in the key for all as the text in the image is small/irregular). On a related note, axis labels and table are a bit small/blurry.
-
All legends, please make clear what data in graphs are, e.g. mean +/- standard deviation. Also, please clarify what the “N” represents whether material replicates, number of gels images with cells, number of cells analyzed, etc.
-
Alamar blue assay axes - while this assay is indicative of viability, in your case, it’s almost used more to assess cell state - proliferative vs. differentiated. Does it make sense to change it to something more general such as “metabolic activity”?
Discussion
-
Line 280 - 282 - include ecm size info for comparison
-
Lines 296 - 299 - discussion of your cells on aligned scaffolds to cells on random scaffolds on glass is not direct. I would review the literature to find alternative studies that examine cell alignment and neuron differentiation on aligned pcl scaffolds
-
Line 305 - 307 - which time point is ChAT first expressed or increased?
-
Line 313 - do you need more markers to confirm Motor neuron identity? Caution against saying they are motor neurons, maybe motor neuron-like?
Methods
-
Line 321 - “1/9 to the gelatin solution” - is this intended to mean the TG is added 1:9 v:v? Could you please clarify this?
-
Line 323 - 2.0 +/- __?
-
Line 352 - how many batches were the 9 images from?
-
LIne 362 - “...laminin to promote cell adhesion”
-
Line 363 - on what day on the scaffolds was differentiation induced?
-
Line 368 - “2 24” is the 2 a typo?
-
Line 385 - consider changing to “the cells on gels”
-
General cell comment - how many cells were seeded onto the gels for the metabolic assay and for imaging? At what point was differentiation induced via media change - before or after cells were seeded onto scaffolds?
Round 2
Reviewer 1 Report
1. Regarding to my former major comment 1, the manuscript got improved with providing the confocal 3D image of the surface topography. However, I am yet convinced by the image, could authors give further quantitative details on the topography, such as the average height of the PCL fiber above the gelatin substrate surface ?. There is a lack of a scale bar for Fig 3a).
2. Regarding to my former major comment 2, I suggest the authors discuss this anisotropic plausibility in the Discussion section and clarify their topographical cue explanation further.
Reviewer 2 Report
Dear authors,
I appreciate your significant edits of the manuscript and find its rigor and clarity are strengthened.
I have a few final edits regarding the labeling of figures that I believe would improve clarity for the readers.
Figures 2/4 - some of the images appear to be low resolution, so please check to make sure they are 300 DPI images for clarity.
Figures 2/4 - please change metabolic viability to metabolic activity.
Figure 3A - consider moving the 3D representations after the brightfield images and add a label similar to the brightfield image labels. I notice that this figure is also in SI - it should only be included in one location.
Figure 5 - is there a reason why the ChAT results were not quantified and included in the table with tubulin? Also, it appears from the representative images in SI that there is a major difference in ChAT expression between random and aligned fibers with aligned fibers promoting more differentiation. This may be worth commenting on in the discussion.
SI figures - I appreciate the addition of the SI figures. They enhance the figure 5 discussion. Will these figures have legends? I don't see the scale for the scale bar?
SI random fiber figure - the 14 day DAPI channel appears to be at a lower magnification than the other images, so I would check to make sure this is the correct image that is paired with these others.
In general, when you switch hydrogel labeling from 15% to 21 kPa, consider labeling 21 kPa (15%), so readers know they are the same hydrogel.
